# Effect of 6-Month Vitamin D Supplementation on Plasma Matrix Gla Protein in Older Adults

**DOI:** 10.3390/nu11020231

**Published:** 2019-01-22

**Authors:** Adriana J. van Ballegooijen, Joline W. J. Beulens, Leon J. Schurgers, Elisa J. de Koning, Paul Lips, Natasja M. van Schoor, Marc G. Vervloet

**Affiliations:** 1Department of Nephrology, and Amsterdam Cardiovascular Science, VU University Medical Center, 1081 HV Amsterdam, The Netherlands; m.vervloet@vumc.nl; 2Department of Epidemiology & Biostatistics, and the Amsterdam Public Health Institute, VU University Medical Center, 1081 HV Amsterdam, The Netherlands; j.beulens@vumc.nl (J.W.J.B.); ej.dekoning@vumc.nl (E.J.d.K.); nm.vanschoor@vumc.nl (N.M.v.S.); 3Julius Center for Health Sciences and Primary Care, University Medical Center Utrecht, 3508 AB Utrecht, The Netherlands; 4Department of Biochemistry, Maastricht University, 6211 LK Maastricht, The Netherlands; l.schurgers@maastrichtuniversity.nl; 5Department of Internal Medicine, and Amsterdam Public Health Research Institute, VU University Medical Center, 1007 MB Amsterdam, The Netherlands; p.lips@vumc.nl

**Keywords:** vitamin D supplementation, vitamin K status, randomized controlled trial

## Abstract

Vitamin D supplementation has been widely promoted to restore 25-hydroxyvitamin D concentrations; however, experimental evidence suggests a nutrient interaction with vitamin K. We assessed the effects of 1200 IU vitamin D_3_ per day versus placebo for six months on vitamin K status in a randomized, double-blind, placebo-controlled trial with participants aged 60–80 years with depressive symptoms and ≥1 functional limitation for a secondary analysis. Stored baseline and six-month follow-up blood samples were available for 131 participants (*n* = 65 placebo vs. *n* = 66 vitamin D supplementation). We measured dephosphorylated uncarboxylated matrix gla protein (MGP) (dp-ucMGP) concentrations—a marker of vitamin K deficiency. Mean age was 68 years, and 89 participants (68%) were women. Vitamin K antagonists were used by 16 participants and multivitamin supplements by 50 participants. No differences in change between intervention and placebo were found (−38.5 ± 389 vs. 4.5 ± 127 (pmol/L), *p* = 0.562). When excluding vitamin K antagonist users and multivitamin users, dp-ucMGP at follow-up was significantly higher in the vitamin D group (*n* = 40) compared to placebo (*n* = 30), with a difference of 92.8 (5.7, 180) pmol/L, adjusting for baseline dp-ucMGP and sex. In conclusion, vitamin D supplementation for six months did not affect vitamin K status; however, among participants without vitamin K antagonist or multivitamin use, vitamin D supplementation influenced dp-ucMGP concentrations.

## 1. Introduction

Vitamin D is a fat-soluble vitamin that can be ingested through foods, such as fatty fish (salmon, mackerel, and tuna), fish liver oils, dairy products, and eggs, but is mainly synthesized by human skin when exposed to sunlight. Vitamin D plays a major role in the regulation of calcium metabolism by increasing intestinal calcium absorption [1]. Over the last few decades, vitamin D supplementation has been promoted to restore 25-hydroxyvitamin D (25(OH)D) concentrations, however, little is known about its potential interaction with other nutrients [2].

Vitamin K is also a fat-soluble vitamin that exists in our diet in two forms: Vitamin K_1_ (phylloquinone), mainly found in green leafy vegetables, such as kale, cabbage, and broccoli, and vitamin K_2_ (menaquinone), mainly found in fermented dairy, such as yogurt and buttermilk, and is produced by lactic acid bacteria in the intestine [3]. Vitamin K is principally transported to the liver, activating coagulation factors; however, it is also transported to extrahepatic tissues, such as the vasculature and bone, regulating the activity of matrix gla protein (MGP) and osteocalcin—two vitamin K-dependent proteins. These proteins require vitamin K for carboxylation in order to exert their functions. In the case of insufficient circulating vitamin K, a greater proportion of MGP and osteocalcin remains uncarboxylated, and is associated with unfavorable outcomes, such as cardiovascular disease, lower bone density, and osteoporosis [4]. The uncarboxylated fraction of osteocalcin and MGP are markers of vitamin K status, and a marker of deficient protection against vascular calcification [5].

Animal and human studies suggest that optimal concentrations of both vitamin D and vitamin K are beneficial for bone and cardiovascular health [6,7,8]. Previous studies have suggested joint associations to vitamin D and K status, with incident hypertension and arterial stiffness among general populations [9,10]. Recent evidence also indicates that vitamin D promotes the production of vitamin K-dependent proteins [11]. This could induce a relative vitamin K deficiency –increased ratio of uncarboxylated over active carboxylated MGP– in case of similar vitamin K intake. Therefore, vitamin D supplementation might increase requirements for vitamin K, however, whether vitamin D supplementation influences vitamin K status is largely unknown.

Only one trial in children studied the effects of vitamin D supplementation on uncarboxylated osteocalcin concentrations, and did not find a difference between vitamin D supplementation and placebo for 12 months [12]. Thus far, the effect of vitamin D supplementation on dephosphorylated uncarboxylated MGP (dp-ucMGP) concentrations has not been investigated. This is of high clinical relevance because it is possible that a relative vitamin K deficiency induced by vitamin D supplementation attenuates protection by activate MGP for vascular calcification.

Therefore, we used data from the D-Vitaal Study, a randomized, double-blind, placebo-controlled trial for a secondary analysis to assess the effects of 1200 IU vitamin D_3_ per day versus placebo, for six months, on dp-ucMGP concentrations.

## 2. Materials and Methods

### 2.1. Study Population

The D-Vitaal Study was a randomized, double-blind, placebo controlled clinical trial. The primary aim was to examine the effects of 1200 IU vitamin D_3_ versus placebo, per day, on depressive symptoms, functional limitations, and physical performance after 12 months, as previously described [13]. For this analysis, we used data from participants with stored baseline and 6-month follow-up blood samples.

The D-Vitaal Study was carried out in Amsterdam and surrounding municipalities in the Netherlands, with participants aged 60–80 years. The majority of participants was recruited through municipality registries. Approximately 20% of participants were recruited through general practitioners (GP) in Amsterdam. Potential participants were screened for low serum 25(OH)D concentrations (≤50 nmol/L in winter (October–March) or 25(OH)D concentrations ≤70 nmol/L in summer (April–September), and the presence of clinically-relevant depressive symptoms and ≥1 functional limitation. Recruitment started in June 2013 and inclusion was finalized in April 2015. In total, 155 participants were randomized. For the present analyses, 6 participants gave no biobank permission and were excluded, and 18 participants had no baseline and follow-up stored blood sample. The analytical sample of our study included 131 participants (Figure 1).

All participants provided written informed consent prior to the start of the intervention. The D-Vitaal Study was approved by the Medical Ethics Committee of the VU Medical Centre, Amsterdam, and was registered in the Netherlands Trial Register: NTR3845.

### 2.2. Measurements

The screening phase of the D-Vitaal Study included a questionnaire, short interview and venipuncture. The screening questionnaire assessed the presence of depressive symptoms, based on a Centre of Epidemiological Studies—Depression Scale (CES-D) score of ≥16, as well as the number of functional limitations. Participants with a major depressive disorder, use of antidepressant medication, a vitamin D supplement dose >400 IU/day, calcium supplementation >1000 mg/day or institutionalized participants were excluded.

### 2.3. Intervention and Randomization

Trained research nurses conducted the interviews and took blood samples at participant homes, at the VU University Medical Center, or at local community medical centers. Participants were randomly allocated at a 1:1 ratio to one of the two blinded treatment groups: Vitamin D_3_ (cholecalciferol) 1200 IU/day or placebo. Participants were stratified by sex, and women were stratified by age (60–70/71–80 years). An independent pharmacist prepared three randomization lists with computer-generated numbers, using blocks of four.

The intervention consisted of three tablets of 400 IU vitamin D_3_, and the placebo group received identical pills without vitamin D, per day. Both the vitamin D and placebo pills were purchased from Vemedia Manufacturing B.V., the Netherlands. Participants were allowed to take a (multi)vitamin D supplement with a maximum of 400 IU/day in addition to the study supplements.

All participants were advised to consume at least three dairy servings daily to ensure an adequate calcium intake of about 1000 mg/day. Calcium intake was assessed with a structured questionnaire during the screening phase. When calcium intake was low (<2 dairy servings per day), 500-mg calcium tablets were prescribed to these participants for daily use.

Participants that discontinued taking study tablets were asked to participate in the remaining follow-up measurements. To encourage study adherence, participants were contacted by phone after two weeks, and three and nine months to check for compliance and adverse events.

### 2.4. Assessment of dp-ucMGP

Dp-ucMGP is a long-term marker of vitamin K status, as opposed to circulating vitamin K_1_ and K_2_, which may fluctuate substantially according to dietary intake [14]. Previously-frozen EDTA plasma samples were stored until December 2017 and shipped to the Department of Biochemistry of Maastricht University in Maastricht to determine dp-ucMGP concentrations. Laboratory personnel was blinded to intervention status. Plasma dp-ucMGP was determined in a single run with a commercially-available IVD CE-marked chemiluminescent InaKtif MGP assay, using the IDS-iSYS system (IDS, Boldon, United Kingdom), blinded for the intervention group. D-Vitaal samples and internal calibrators were incubated with magnetic particles coated with murine monoclonal antibodies, dpMGP; acridinium-labelled murine monoclonal antibodies, ucMGP; and an assay buffer. The magnetic particles were captured using a magnet and washed to remove any unbound analytes. Thereafter, trigger reagents were added and the resulting light emitted by the acridinium label was directly proportional to the level of dp-ucMGP in the sample. The within-run and total variations were 0.8–6.2% and 3.0–8.2%, respectively.

### 2.5. Other Variables

D-Vitaal study personnel collected information from participants regarding age, sex, and medication use, including vitamin D and multi-vitamin use, at the time of study entry. The date of drawing blood was used to define the summer season (April–September). Study compliance was checked by tablet count and divided into “lower” or “higher” than 80%.

The Endocrine Laboratory of the VU University Medical Center conducted the 25(OH)D determinations. Participants were fasted with regard to dairy products. Serum 25(OH)D was determined using liquid chromatography followed by tandem mass spectrometry.

### 2.6. Statistical Analyses

We tabulated baseline characteristics for vitamin D supplementation use and placebo according to means and standard deviation (SD), or number and percentage. Further, we summarized baseline, follow-up and delta (follow-up minus baseline) dp-ucMGP concentrations by means and SD.

We used the independent-samples *t*-test to estimate the difference in change between the vitamin D and placebo group. Further, we performed linear regression analysis using follow-up dp-ucMGP as outcome, adjusting for baseline dp-ucMGP and potential confounders. When large differences in baseline characteristics occurred (>10%) between the groups, we adjusted for baseline characteristics. 

A sensitivity analysis was performed, excluding factors that might influence vitamin K metabolism by excluding vitamin K antagonist or multivitamin use (usually between 10–80 µg vitamin K). Further, we restricted this sensitivity analysis to participants with compliance ≥80%.

We conducted analyses using SPSS, version 22 (SPSS Inc., Chicago, IL, USA) and considered a 2-sided *p* < 0.05 to be statistically significant.

## 3. Results

### 3.1. Study Population

Among the 131 D-Vitaal participants, the mean age was 68 years SD (5.2) and 89 (68%) were women. Vitamin K antagonists were used by 16 participants, multivitamins by 50 participants, and four participants used a vitamin D supplement in addition to the intervention tablets. No major differences in baseline characteristics were observed between the vitamin D and placebo groups (Table 1).

When excluding vitamin K antagonist and multivitamin users (*n* = 61), the vitamin D group contained a higher percentage of women (58%) than placebo (47%). Of the 16 multi-vitamin users, four of them also used a vitamin D supplements and were excluded. No other notable differences in baseline characteristics were observed between the intervention groups. Figure 2 depicts a baseline scatter plot of plasma 25(OH)D and dp-ucMGP by multivitamin supplement use, with a slightly positive trend for higher 25(OH)D and higher dp-ucMGP, although the number of background multivitamin users was small (*n* = 50).

### 3.2. Effect Vitamin D Supplementation

Mean dp-ucMGP concentrations were slightly higher in the placebo group at baseline and at follow-up: 735 ± 638 and 697 ± 609 pmol/L in the placebo group vs. 651 ± 462 and 655 ± 438 pmol/L in the vitamin D group (Table 2). The dp-ucMGP concentrations were slightly skewed to the right, but the residuals of dp-ucMGP were approximately normally distributed and therefore not transformed.

The linear regression analysis to test the effect of vitamin D supplementation on dp-ucMGP concentrations adjusting for baseline dp-ucMGP and sex demonstrated that there was no significant difference between placebo and intervention at end of the study (29.5 (95% CI −64.5, 123) pmol/L).

### 3.3. Sensitivity Analysis

When excluding vitamin K antagonist and multivitamin users, the mean dp-ucMGP concentrations for baseline and follow-up in the placebo group (*n* = 30) were: 596 ± 355 and 504 ± 171 pmol/L vs. 577 ± 311, and 580 ± 277 pmol/L in the vitamin D group (*n* = 40). More women were present in the vitamin D group (*n* = 23) (58%) than the placebo group (*n* = 14) (47%). The dp-ucMGP at follow-up was significantly higher in the vitamin D group compared to the placebo group: 92.8 (5.7, 180) pmol/L, *p* = 0.034 adjusting for baseline dp-ucMGP and sex. Natural log transformed data yielded similar results. Further restricting our sample to participants with a supplement compliance >80% (*n* = 64) resulted in a more pronounced difference: 111 (16.6, 205) pmol/L, *p* = 0.022.

## 4. Discussion

In this randomized double-blind placebo-controlled study, we assessed the effect of 1200 IU vitamin D_3_ supplementation per day versus placebo for six-months on dp-ucMGP concentrations in participants with clinically-relevant depressive symptoms and ≥1 functional limitation, between 60–80 years of age, as a secondary analysis. No significant differences in terms of change in dp-ucMGP between the vitamin D and placebo group was found after a six-month follow-up. In a sensitivity analysis, in which we excluded vitamin K antagonist and multivitamin users, the difference in dp-ucMGP concentrations between the vitamin D supplement group and the placebo group was significant. This may imply that vitamin D supplementation affects vitamin K status.

This is the first study to investigate the effect of vitamin D supplementation on dp-ucMGP concentrations, the vascular marker of vitamin K deficiency. One study investigated the effect of vitamin D supplementation on osteocalcin, a vitamin K dependent protein, in girls. Among 67 Danish girls (aged 11–12 years), no effect was found after 12-months daily supplementation with 10 µg (400 IU) vitamin D_3_ vs. placebo on serum percentage of undercarboxylated osteocalcin [12]. This could be due to sexual maturity of the girls over 12 months or increased nutrient requirements, which makes it hard to study nutrient status.

In a cross-sectional study among Italian hemodialysis patients, treatment with vitamin D analogs (20%) was associated with a higher percentage total and uncarboxylated osteocalcin concentrations; however, no significant association with total and uncarboyxlated MGP was observed [15].

A study among Japanese women observed that vitamin K_2_ and 1α-hydroxyvitamin D_3_ alfacalcidol (vitamin D receptor activator) vs. vitamin K_2_ significantly decreased the percentage of uncarboxylated osteocalcin concentrations after two years, and also the total osteocalcin significantly decreased in the vitamin K_2_ and alfacalcidol groups, suggesting a better active osteocalcin ratio [16]. In summary, this Japanese study indicates a potential effect of vitamin D on vitamin K-dependent proteins.

### 4.1. Potential Mechanisms of Vitamin D Supplementation on Vitamin K Status

The MGP-gene promoter contains a vitamin D response element, capable of a two- to three-fold enhanced MGP expression after vitamin D binding [6,17]. In an in vitro study, excess vitamin D induced a relative vitamin K deficiency by direct stimulation of the synthesis of vitamin K-dependent proteins [6]. Treatment with 1,25(OH)_2_D dramatically increased MGP mRNA and increased MGP secretion 15-fold [18]. The increased production of vitamin K dependent proteins, increases vitamin K requirements. Vitamin D supplementation might therefore increase the ratio of uncarboxylated over active carboxylated MGP.

Some animal studies confirmed that vitamin D has direct effects on vitamin K dependent metabolism [6,7,19,20]. In mice, a diet high in 1,25(OH)_2_D for 20 weeks resulted in higher amounts of uncarboxylated MGP and renal calcification [19]. In rats, vitamin K, in turn, suppressed soft tissue calcification induced by vitamin D as demonstrated by lower calcium and phosphorus contents in the aorta and kidney [7]. In summary, emerging evidence suggests that vitamin D supplementation can increase vitamin K requirements.

### 4.2. Clinical Implications

Worldwide, a large group of people is prescribed to a supplemental regime of vitamin D. In Europe, depending on country and sex, vitamin D supplementation, often between 400–800 IU, is used by approximately 30% of the adult population [21,22]. Large amounts of vitamin D supplementation are generally considered safe, with doses up to 4000 international units (IU) per day [3]; however, little is known about the potential effects and side-effects of long-term high-dose vitamin D supplementation [23]. Our findings question the safety of high vitamin D supplementation (12,000 IU/day) in the setting of vitamin K deficiency. Adverse effects of prolonged vitamin D supplementation of even low dose vitamin D supplements (up to 400 IU/day) have been reported with greater risks of hypercalciuria, kidney stones, hypercalcemia, and vascular lesions [23,24,25,26,27,28]. Furthermore, multiple meta-analyses of randomized controlled trials even question the efficacy of vitamin D supplementation for musculo-skeletal health, and fractures [28,29,30,31]; however, the meta-analysis of Bolland et al. [29] excluded studies with combined vitamin D and calcium supplementation and included studies with a follow-up time of six months, which is insufficiently long for fracture outcomes. Nonetheless, these studies concluded that there is little justification to use vitamin D supplements to maintain or improve health outcomes.

More clinical data about the potential interplay between vitamin D and vitamin K metabolism are urgently needed. Multiple lines of evidence suggest that optimal concentrations of both vitamin D and vitamin K might protect against progressive vascular calcification and cardiovascular disease [11]. The combination of vitamin D and K supplementation might overcome the negative effects of vitamin D on vitamin K metabolism, but this should be tested in clinical studies.

### 4.3. Strengths and Limitations

The strengths of our study include the randomized, double-blind, placebo-controlled design, good compliance and a low drop-out rate. The study included vitamin D deficient participants with a low baseline 25(OH)D concentration (≤50 nmol/L in winter or 25(OH)D ≤ 70 nmol/L in summer). This allowed us to study the effects of vitamin D restoration on vitamin K status.

There were some limitations that should be noted as well. This is a secondary analysis, since this study was not designed to investigate the effect of vitamin D supplementation on vitamin K status. Due to the exclusion of vitamin K antagonist users and multivitamin users, randomized participants were excluded and the vitamin D group contained a higher percentage of women than placebo. Therefore, the results should be interpreted with caution and are hypothesis generating. Further, circulating MGP concentrations do not necessarily reflect MGP at tissue concentrations of the vasculature. Therefore, we can only speculate how circulating MGP as vascular marker of vitamin K status, is related to tissue MGP and can thereby potentially lower vascular calcification. However, previous studies did find associations with circulating dp-ucMGP and vascular calcification, underlining validity of our approach.

Baseline 25(OH)D concentrations were indicative for mild vitamin D deficiency, which may have limited the effect of vitamin D supplementation. In addition, participants were allowed to use low dose vitamin D supplements. When those subjects were excluded along with multivitamin users and vitamin K antagonist users, we did find an effect of vitamin D supplementation on vitamin K status. It should be noted that this was in a sensitivity analysis and the vitamin D group contained a higher percentage of women (58% vs. 47%). Lastly, included participants had clinically-relevant depressive symptoms and ≥1 functional limitation, which limited the generalizability to the general population.

## 5. Conclusions

In conclusion, daily supplementation with 1200 IU of vitamin D_3_ for six months did not affect vitamin K status in adults with mild vitamin D deficiency and depressive symptoms and ≥1 functional limitation. However, among participants without vitamin K antagonist use and multivitamin use, vitamin D supplementation increased dp-ucMGP concentrations after six months compared to a placebo. This may imply that vitamin D supplementation may induce a relative vitamin K deficiency and may diminish the protection by active MGP against vascular calcification. However, further studies are needed to elucidate whether vitamin D supplementation can influence vitamin K status.

## Figures and Tables

**Figure 1 nutrients-11-00231-f001:**
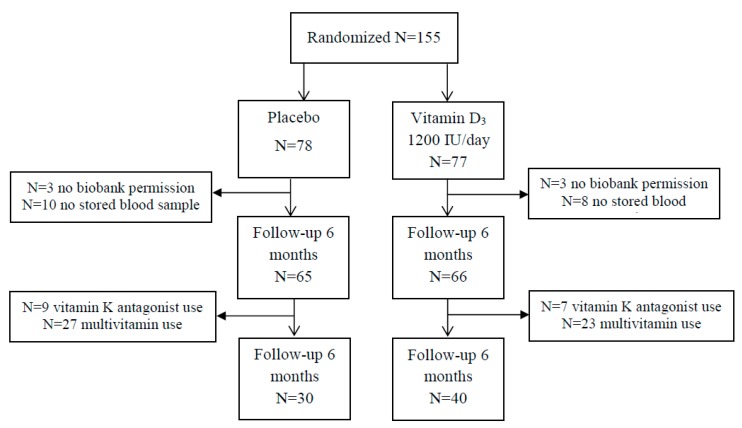
Flow chart of randomization. Due to combined use of vitamin K antagonist use and multivitamin use, the numbers are higher in the final population (*n* = 1, placebo; *n* = 4, vitamin D group).

**Figure 2 nutrients-11-00231-f002:**
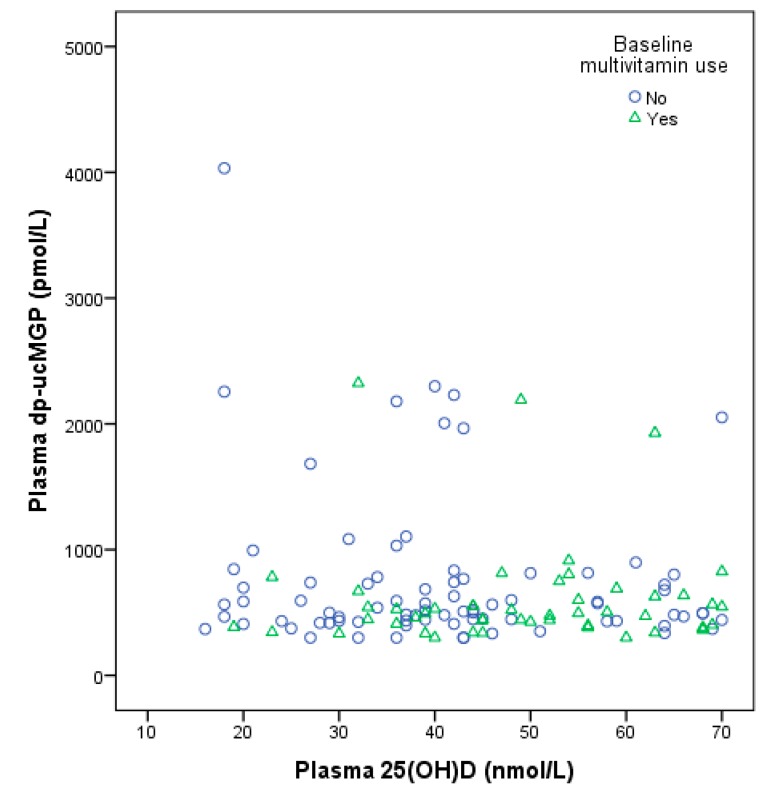
Baseline scatter plot of plasma 25-hydroxvitamin D and plasma dp-ucMGP concentrations by multivitamin use. Rounds: No multivitamin, *n* = 81; triangles: Multivitamin use, *n* = 50.

**Table 1 nutrients-11-00231-t001:** Baseline characteristics of D-Vitaal study participants.

	Placebo *N* = 65	Vitamin D_3_ *N* = 66	Placebo *N* = 30 **	Vitamin D_3_ *N* = 40 **
Female sex (%)	44 (56%)	45 (58%)	14 (47%)	23 (58%)
Age (year)	67.7 ± 5.1	68.5 ± 5.3	67.1 ± 4.9	68.5 ± 5.1
Baseline summer season	38 (59%)	37 (56%)	16 (53%)	18 (45%)
Plasma 25(OH)D (nmol/L)	43.3 ± 14.5	45.1 ± 15.3	40.4 ± 14.5	42.9 ± 15.2
Plasma dp-ucMGP (pmol/L)	735 ± 638	651 ± 462	596 ± 355	504 ± 171
Vitamin K antagonist use	9 (14%)	7 (11%)	-	-
Multivitamin use	27 (44%)	23 (35%)	-	-
Vitamin D use *	1 (2%)	3 (5%)	-	-
Compliance > 80%	59 (91%)	62 (94%)	27 (90%)	38 (95%)

Values represent number and percentages or mean and standard deviation. * Vitamin D use by participant max 400 IU vitamin D_3_. ** excluding vitamin K antagonist users and multivitamin use.

**Table 2 nutrients-11-00231-t002:** dp-ucMGP concentrations at baseline and 6-month follow-up by intervention groups.

	*N*	Baseline dp-ucMGP (pmol/L)	6-month dp-ucMGP (pmol/L)	Delta dp-ucMGP (pmol/L)	*p*-Value
All study participants					
Placebo	65	735 ± 638	697 ± 609	−38.5 ± 389	0.562
Vitamin D_3_	66	651 ± 462	655 ± 438	4.5 ± 127	
Adjusted result				29.5 (−64.5, 123)	0.536
No multivitamin and VKA					
Placebo	30	596 ± 355	504 ± 171	−91.9 ± 359	0.125
Vitamin D_3_	40	577 ± 311	580 ± 277	3.0 ± 125	
Adjusted result				92.8 (5.7, 180)	0.034

Values are mean and standard deviation or regression coefficients and 95% confidence adjusted for baseline dp-ucMGP and sex intervals; dp-ucMGP: dephosphorylated uncarboxylated matrix gla protein, VKA: vitamin K antagonist Natural log transformed data yielded similar results.

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
