# Peer review of "Effect of 6-Month Vitamin D Supplementation on Plasma Matrix Gla Protein in Older Adults"

_nutrients, 2019, doi:10.3390/nu11020231_

Round 1
Reviewer 1 Report
line 25: Mean age was 68 years and 68% (89) of the participants were women. Please change
line 26:Vitamin K antagonists were used by 16 participants and multivitamins were used by 50 participants. Please change
line 201: please check if 10 ug or 10mg vitamin D.
The rest of the article looks fine.
Author Response
We thank the editor and referees for carefully reading our manuscript and their valuable comments. We highlighted changes in the manuscript and provide our responses to the reviewers’ questions and concerns below. Both reviewers underscored the novelty and the importance of the study.
Reviewer 1
line 25: Mean age was 68 years and 68% (89) of the participants were women. Please change
Thanks for spotting this error. We placed the percentage sign behind the mean age within brackets on page 2: Mean age was 68 years and 89 (68%) were women.
line 26: Vitamin K antagonists were used by 16 participants and multivitamins were used by 50 participants. Please change
Thank you for the suggestion. We have changed this accordingly and now read as: “Vitamin K antagonists were used by 16 participants and multivitamin supplements by 50 participants”.
line 201: please check if 10 ug or 10mg vitamin D.
We have corrected this mistake and use 10µg vitamin D throughout the manuscript.
The rest of the article looks fine.
Thank you for the valuable suggestions.
Reviewer 2 Report |
This small study shows that vitamin D supplementation might affect vitamin K status. Although large dose of vitamin D supplementation is claimed to be safe, this study showed that 1200 IU vitamin D3 per day for 6 months could induce a relative vitamin K deficiency & may generate a procalcifying microenvironment to facilitate vascular lesions. The results are convincing with potential clinical implications. I've following suggestion:
In the discussion section, authors might consider adding a section on “Potential adverse effects of prolonged vitamin D supplementation” & discuss the results of the recent publications, showing no measurable effects of vitamin D supplements (Lancet Diabetes Endocrinol. 2018 Nov;6(11):847-858; 180:81-86; N Engl J Med. 2018 Nov 10. doi: 10.1056/NEJMoa1809944; N Engl J Med. 2018 Aug 9;379(6):535-546.); potential complications of long-term consumption of supplements should be discussed (J Steroid Biochem Mol Biol. 2018 Jun;180:81-86). The readers should be informed that even so-called safe dose of vitamin D supplements might not be safe, & could induce vascular pathology & beyond.
Author Response
Rebuttal Nutrients
Manuscript ID: nutrients-411605
Effect of 6-month vitamin D supplementation on plasma matrix gla protein in older adults
We thank the editor and referees for carefully reading our manuscript and their valuable comments. We highlighted changes in the manuscript and provide our responses to the reviewers’ questions and concerns below. Both reviewers underscored the novelty and the importance of the study.
Comments and Suggestions for Authors |
Reviewer 2
This small study shows that vitamin D supplementation might affect vitamin K status. Although large dose of vitamin D supplementation is claimed to be safe, this study showed that 1200 IU vitamin D3 per day for 6 months could induce a relative vitamin K deficiency & may generate a procalcifying microenvironment to facilitate vascular lesions. The results are convincing with potential clinical implications. I've following suggestion:
In the discussion section, authors might consider adding a section on “Potential adverse effects of prolonged vitamin D supplementation” & discuss the results of the recent publications, showing no measurable effects of vitamin D supplements (Lancet Diabetes Endocrinol. 2018 Nov;6(11):847-858; 180:81-86; N Engl J Med. 2018 Nov 10. doi: 10.1056/NEJMoa1809944; N Engl J Med. 2018 Aug 9;379(6):535-546.); potential complications of long-term consumption of supplements should be discussed (J Steroid Biochem Mol Biol. 2018 Jun;180:81-86). The readers should be informed that even so-called safe dose of vitamin D supplements might not be safe, & could induce vascular pathology & beyond.
Thank you for raising this point. We have added a paragraph about the potential adverse effects of prolonged vitamin D supplementation and included recent literature. We further stressed that little is known about potential long-term effects of vitamin D supplementation and adverse effects of even low dose vitamin D supplements have been reported. Discussion page 13, first paragraph:
Adverse effects of prolonged vitamin D supplementation of even low dose vitamin D supplements (up to 400 IU/day) have been reported with greater risks of hypercalciuria, kidney stones, hypercalcemia, and vascular lesions [23-28]. Furthermore, multiple meta-analyses of randomized controlled trials even question the efficacy of vitamin D supplementation for musculo-skeletal health, and fractures [28-31], however, the meta-analysis of Bolland et al [29] excluded studies with combined vitamin D and calcium supplementation and included studies with a follow-up time of 6 months, which is insufficiently long for fracture outcomes. Nonetheless, these studies conclude that there is little justification to use vitamin D supplements to maintain or improve health outcomes.”